# WHAT INFORMATION DOES A RESNET COMPRESS?

## ABSTRACT

The information bottleneck principle (Shwartz-Ziv & Tishby, 2017) suggests that SGD-based training of deep neural networks results in optimally compressed hidden layers, from an information theoretic perspective. However, this claim was established on toy data. The goal of the work we present here is to test whether the information bottleneck principle is applicable to a realistic setting using a larger and deeper convolutional architecture, a ResNet model. We trained PixelCNN++ models as inverse representation decoders to measure the mutual information between hidden layers of a ResNet and input image data, when trained for (1) classification and (2) autoencoding. We find that two stages of learning happen for both training regimes, and that compression does occur, even for an autoencoder. Sampling images by conditioning on hidden layers' activations offers an intuitive visualisation to understand *what a ResNets learns to forget*.

## 1  INTRODUCTION

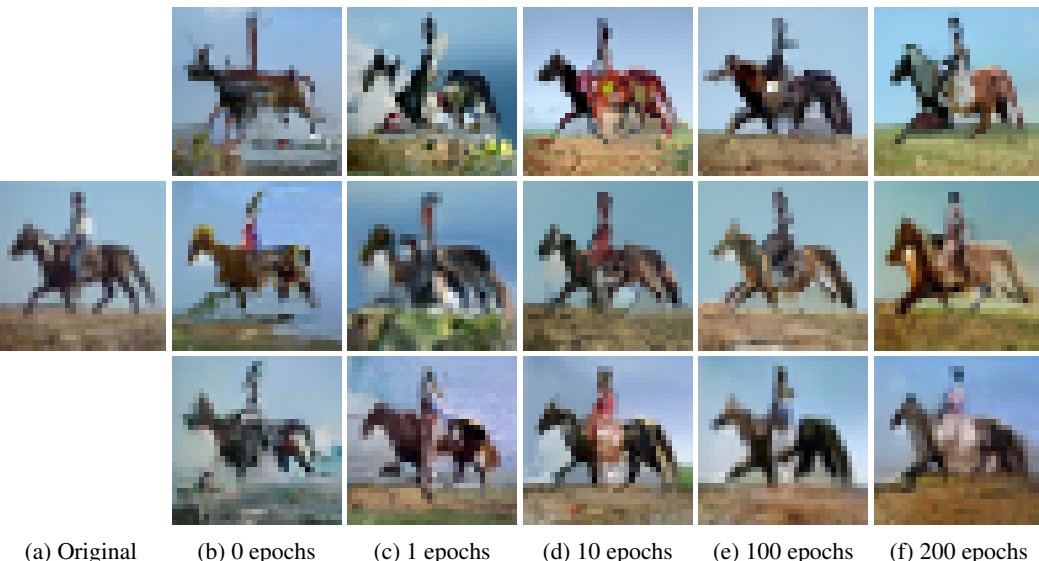

|                |              |             |              |                |              |
|----------------|--------------|-------------|--------------|----------------|--------------|
| (a) Original   | (b) 0 epochs | (c) 1 epochs | (d) 10 epochs | (e) 100 epochs | (f) 200 epochs |

Figure 1: Samples generated using a PixelCNN++ decoder model, conditioned on hidden activations created by processing an image of a horse (a) in a ResNet ($\mathbf{h}_3$, see Section 5) in classifier training. Conditionally generated images are shown in (b) - (f). Ten epochs is the *peak of fitting*, and 200 epochs is the *end of compression*. These samples enable an intuitive illustration of compression in hidden layers. Based on this example it seems that a compressed representation (f) results in varied samples because it compresses class-irrelevant information. Compare the beginning (d) to the end (f) of compression: there is greater variety at the end without losing the essence of 'horse'.

Deep neural networks are ubiquitous in machine learning and computer vision. Unfortunately, the popularity of neural networks for applications is unmatched by an agreed upon and clear understanding of how exactly they work, and why they generalise well. The field of deep learning will advance with more comprehensive theory and empirical studies that better characterise neural networks.

Generalisation, in the context of learning, means the extraction of typical abstract properties of training data that can be successfully used for inference on unseen data. Neural networks generalise well even when they are over-parameterised. Zhang et al. (2017) highlighted the need to rethink generalisation, because conventional wisdom is not readily applicable to neural networks. A number of avenues for new generalisation bounds for neural networks (Bartlett et al., 2017; Golowich et al., 2017; Neyshabur et al., 2017) exemplify how inapplicable conventional methods for understanding model generalisation can be.

One approach to better understanding neural networks is the Information Bottleneck (IB, Section 2) interpretation of deep learning (Tishby et al., 2000; Tishby & Zaslavsky, 2015; Shwartz-Ziv & Tishby, 2017). The IB accredits deep learning success to compression in hidden layers via the noise of parameter updates in stochastic gradient descent (SGD). Information compression results in *optimal representations that discard task-irrelevant data while keeping task-relevant data*.

The IB principle has since been actively debated (Saxe et al., 2018), partially motivating this work. The novelty is that we apply information theoretic analyses to **modern convolutional residual neural networks** (ResNets, Section 5) trained on **realistic images** (Section 5.2). These choices complicate analysis since information quantities are non-trivial to compute for high-dimensions. Our solution is to define a lower bound on the mutual information (MI, Section 4) and to estimate it by training decoder models (Section 4.1). The decoder model for the MI between images and hidden layers is a conditional PixelCNN++ and samples generated therefrom illustrate visually MI (Figure 1).

## 1.1 OUR CONTRIBUTIONS

- An experimental framework for tracking the MI in a realistic setting. Tracking both the forward and inverse MI in a ResNet using realistic images. Earlier research tracked these quantities for constrained toy-problems or low-dimensional data. Lifting these constraints requires defining models to compute a lower bound on the MI.
- Analysis of PixelCNN++ samples conditioned on hidden layer activations to illustrate the *type of information* that ResNet classifiers learn to compress. This is done via the visual demonstration of the sorts of invariances that a ResNet learns.

This paper compliments earlier work on the IB interpretation of deep learning, which is described in the next section. The key difference is that we analyse a modern network trained on realistic images.

## 2 INFORMATION BOTTLENECK INTERPRETATION OF DEEP LEARNING

The IB interpretation of learning (Tishby & Zaslavsky, 2015; Shwartz-Ziv & Tishby, 2017) suggests that an optimal representation exists between input data, $\mathsf{x}$, and target data, $\mathsf{y}$, that captures all relevant components in $\mathsf{x}$ about $\mathsf{y}$. An optimal representation, $\mathsf{h}$, should retain *only* the information relevant for the task described by $\mathsf{y}$.

The IB interpretation posits that the hidden layers in neural networks learn hidden layer configurations that maximally compress characteristics in the input data that are irrelevant to the target task. In classification, for example: the nature of the ground and/or background in an image of a horse may not matter and could be discarded, provided the horse remains (see Figure 1).

**Mutual Information**    We interpret the activations of hidden layers as random variables so that we can compute and track the mutual information (MI) between these layers and data. MI is defined as:

$$I(\mathsf{x}; \mathsf{y}) = \int_{\mathsf{y}} \int_{\mathsf{x}} p(\mathsf{x}, \mathsf{y}) \log \frac{p(\mathsf{x}, \mathsf{y})}{p(\mathsf{x})p(\mathsf{y})} \, d\mathsf{x} \, d\mathsf{y}, \tag{1}$$

which is the Kullback-Leibler (KL) divergence between the joint distribution of two random variables and the product of their marginals.

**Application to Deep Learning**    Shwartz-Ziv & Tishby (2017) applied the IB principle to deep learning. By studying what they called the information plane – how $I(\mathsf{x}; \mathsf{h})$ and $I(\mathsf{y}; \mathsf{h})$ changed

over time (h is a hidden layer, x is input data, and y is the target) – they showed that neural networks have two learning phases:

1. *Fitting*, or empirical error minimisation, where the information shared between hidden representations and input data was maximised.
2. *Compression*, where the the information shared between hidden representation and input data was minimised but constrained to the classification task at hand.

We refer the reader to Sections 2.1 to 2.3 in Shwartz-Ziv & Tishby (2017) for more detail on the information plane analysis. Generalisation via compression was put forward as the reason deep learning is successful. From the IB perspective, an advantage of depth is computational in that it shortens compression time.

## 3 RELATED WORKS

Applying the IB principle to deep learning goes some way to give a theoretical explanation of why neural networks generalise well. However, empirical research to determine its true relevance and applicability is paramount – we contribute here by analysing a modern network using realistic data to see if the principle of compression for generalisation holds in this case.

**On the IB interpretation of deep learning**   Saxe et al. (2018) constructed several experiments to explore and refute the IB interpretation. They demonstrated:

1. The choice of non-linearity with a binning methodology for MI computation may explain the compression behaviour. Shwartz-Ziv & Tishby (2017) used a *saturating non-linearity* that, when paired with a binned MI computation, seems to confound whether compression occurs. Saxe et al. (2018) used the non-saturating ReLU and did not see compression.
2. Generalisation does not require compression. A deep linear network toy example was constructed to show generalisation without compression.
3. Stochastic relaxation is not the mechanism of compression. Neural networks with ReLU activations did not compress (according to the binned MI calculation) but still switched from high to low SNRs.

What we seek to show in this paper is that modern convolutional networks do evidence information compression during training. We use the ReLU family of non-linearities and instead of binning to compute MI, we use decoding models. Therefore, our experiments aim to contextualise further the IB principle and its applicability to deep learning theory.

**Reversible Networks**   Jacobsen et al. (2018) queried whether compression is necessary for generalisation by constructing an invertible convolutional neural network (CNN). They posited that a reversible network is an example that refutes the IB principle because information is never discarded. They also provided an accompanying counter explanation: where depth is responsible for progressively separating data (in the inter-class sense) and contracting data (in the intra-class sense). Although intentionally reversible networks (Jacobsen et al., 2018) do not discard irrelevant components of the input space, we postulate that these instead *progressively separate the irrelevant components from the relevant*, allowing the final classification mapping to discard this information.

**Inverting Supervised Representations**   Concurrent to the work in this paper, Nash et al. (2018) trained conditional PixelCNN++ models to 'invert' representations learned by a CNN classifier. Using the MNIST (Lecun et al., 1998) and CIFAR-10 (Krizhevsky, 2009) image datasets, they showed how a PixelCNN++ can be used as a tool to analyse the invariances and behaviour of hidden layers.

The MI was tracked using a PixelCNN++ model, as here. However, the model tested was much smaller than the ResNet we inspect here; we test both classification and autoencoding, whereas that work only considered classification; we provide a finer granularity of assessment, including at model initialisation; and the conditionally generated samples we provide illustrate greater variation owing to the deeper ResNet.

In the next section we present and discuss a lower bound on the MI.

## 4   MUTUAL INFORMATION LOWER BOUND COMPUTATION

Earlier work applied various binning strategies to toy data. Binning is not applicable to the modern networks we study here because images as input have many more dimensions than toy datasets. We derive a lower bound, similar to Alemi et al. (2017), on the MI between two random vectors as follows:

$$
I(\mathbf{x}; \mathbf{h}) = \mathbb{E}_{p_D(\mathbf{x},\mathbf{h})} \left[ \log \frac{p_D(\mathbf{x} \mid \mathbf{h}) q(\mathbf{x} \mid \mathbf{h})}{q(\mathbf{x} \mid \mathbf{h})} \right] - C
$$

$$
= \mathbb{E}_{p_D(\mathbf{x},\mathbf{h})} \left[ \log q(\mathbf{x} \mid \mathbf{h}) \right] + \mathbb{E}_{p_D(\mathbf{h})} \left[ D_{KL}(p_D(\mathbf{x} \mid \mathbf{h}) || q(\mathbf{x} \mid \mathbf{h})) \right] - C
$$
$$
\geq \mathbb{E}_{p_D(\mathbf{x},\mathbf{h})} \left[ \log q(\mathbf{x} \mid \mathbf{h}) \right] - C,
$$

(2)

where $\mathbf{x}$ is the input image data, $\mathbf{h}$ is a hidden representation (the activations of a hidden layer), $p_D$ is the true data distribution, and $q$ is an auxiliary distribution introduced for the lower bound. We need not estimate $C = \mathbb{E}_{p_D(\mathbf{x})} \left[ \log p_D(\mathbf{x}) \right]$, since the entropy of the data is constant. The lower bound follows since the KL-divergence is positive. We can replace $\mathbf{x}$ with $\mathbf{y}$ for the analogous quantity w.r.t. the target data. With sufficiently large data (of size N) we can estimate:

$$
\mathbb{E}_{p_D(\mathbf{x},\mathbf{h})} \left[ \log q(\mathbf{x} \mid \mathbf{h}) \right] \simeq \frac{1}{N} \sum_{\mathbf{x}^{(i)}, \mathbf{h}^{(i)}} \log q(\mathbf{x}^{(i)} \mid \mathbf{h}^{(i)}),
$$

(3)

where $\mathbf{x}^{(i)}$ and $\mathbf{h}^{(i)}$ are images and the activations of a hidden layer, respectively. The task now becomes defining the decoder models $q(\mathbf{x} \mid \mathbf{h})$ and $q(\mathbf{y} \mid \mathbf{h})$ to estimate the MI.

### 4.1   DECODER MODELS

MI is difficult to compute unless the problem and representations are heavily constrained or designed to make it possible. We do not enforce such constraints, but instead define decoder models that can estimate a lower bound on the MI.

**Forward direction: computing $I(\mathbf{y}; \mathbf{h})$**   The decoder model for $q(\mathbf{y} \mid \mathbf{h})$ is constructed as a classifier model that takes as input a hidden layer's activations. To simplify matters, we define this decoder model to be identical to the encoder architecture under scrutiny (Section 5). To compute the MI between any chosen hidden representation, $\mathbf{h}_j$ (where $j$ defines the layer index), we freeze all weights prior to this layer, reinitialise the weights thereafter, and train these remaining weights as before (Section 5.1).

**Inverse direction: computing $I(\mathbf{x}; \mathbf{h})$**   The input images – $\mathbf{x} \in \mathbb{R}^{M \times M \times 3}$, where $M = 32$ is the image width/height – are high-dimensional. This makes estimating $I(\mathbf{x}; \mathbf{h})$ more complicated than $I(\mathbf{y}; \mathbf{h})$. To do so, we use a PixelCNN++ model (Salimans et al., 2017): a state-of-the-art autoregressive explicit density estimator that allows access to the model log-likelihood (a critical advantage over implicit density estimators). See Appendix B for more details.

**A note on the quality of the lower bound**   The tightness of the lower bound is directly influenced by the quality of the model used to estimate it. We take a pragmatic stance on what sort of error to expect: using a PixelCNN++ to decode $I(\mathbf{x}; \mathbf{h})$ essentially estimates the level of *usable* information, in as much as it can recover the input images. A similar argument can be made for the forward direction, but there is no direct way of measuring the tightness of the bound. Even though empirical research such as this could benefit from an ablation study, for instance, we leave that to future work.

## 5   EXPERIMENTAL PROCEDURE

The architecture used to encode hidden layers was taken directly from the original ResNet (He et al., 2016) classifier architecture for CIFAR-10. This is either trained for classification or autoencoding. Further, this architecture is not-altered when using it to do forward decoding (Section 4.1).

We define four hidden layers for which we track the MI: $\mathbf{h}_1$, $\mathbf{h}_2$, $\mathbf{h}_3$, and $\mathbf{h}_4$. We *sample from these* (Equation 3) as the activations: at the end of the three hyper-layers of the ResNet ($\mathbf{h}_1$, $\mathbf{h}_2$, and $\mathbf{h}_3$); and $\mathbf{h}_4$ after a $4 \times 4$ average pooling of $\mathbf{h}_3$ (see Figure 6 in Appendix A). $\mathbf{h}_4$ is the penultimate layer and is therefore *expected to be most compressed*. None of these layers have skip connections over them. Regarding the autoencoder's decoder, this is created by simply inverting the architecture using upsampling. The autoencoder's bottleneck is $\mathbf{h}_4$.

The hyper-parameters for the PixelCNN++ decoder models were set according to the original paper. Regarding conditioning on $\mathbf{h}$: this is accomplished by either up- or down-sampling $\mathbf{h}$ to fit all necessary layers (Appendix B.1 expands on this).

## 5.1 TRAINING

Both the classifier and autoencoder weights were optimised using SGD with a learning rate of $0.1$ and cosine annealing to zero over 200 epochs, a momentum factor of $0.9$ and a L2 weight decay factor of $0.0005$. We used the leaky ReLU non-linearity. Cross-entropy loss was used for the classifier, while mean-squared-error (MSE) loss was used for the autoencoder. Our implementation was written in PyTorch (Paszke et al., 2017). For clarity, Algorithm 1 in Appendix C gives a breakdown of the experimental procedure we followed.

## 5.2 DATA: CINIC-10

The analysis in this paper requires computing MI using decoder models, presenting a challenge in that this is a *data-hungry* process. We need: (1) enough data to train the models under scrutiny; (2) enough data to train the decoder models; and (3) enough data for the actual MI estimate (Equation 3). Moreover, the above requirements need *separate data drawn from the same distribution* to avoid data contamination and overfitting, particularly for PixelCNN++. Hence, we require a three-way split:

1. **Encoding**, for training the autoencoder and classifer;
2. **Decoding**, for training the models under which MI is computed; and
3. **Evaluation**, to provide unbiased held-out estimates of $I(\mathbf{x}; \mathbf{h})$ and $I(\mathbf{y}; \mathbf{h})$.

Since CIFAR-10 (Krizhevsky, 2009) is too small and Imagenet (Deng et al., 2009) is too difficult, we used a recently compiled dataset called CINIC-10: CINIC-10 is Not Imagenet or CIFAR-10 (Darlow et al., 2018). It was compiled by combining (downsampled) images from the Imagenet database with CIFAR-10. It consists of 270,000 images split into 3 equal subsets, which we use as the encoding, decoding, and evaluation sets. In the next section we discuss observations from tracking MI.

## 6 OBSERVATIONS: IB PRINCIPLE FOR A RESNET

Shwartz-Ziv & Tishby (2017) made a number of claims regarding deep learning. We make observations in this section and connect them to the IB interpretation. In Section 6.1 we show and discuss a series of figures that shows that both fitting and compression occur in a ResNet. In Section 6.2 we illustrate the quality of information kept and discarded by analysis of conditionally generated images from the PixelCNN++ model.

### 6.1 MI TRACKING

Figure 2 gives the information curves expressed in the same fashion as in earlier work (Shwartz-Ziv & Tishby, 2017; Saxe et al., 2018); Figure 3 tracks the MI for classifier and autoencoder training. Appendix E gives some training curves for the PixelCNN++ decoding models to show their convergence behaviour and clarify the associated computational burden of this work.

**Classification** For classification $I(\mathbf{y}; \mathbf{h}_j)$ always increases and greater changes in MI can be seen for later layers. The convergence point (200 epochs) is the point at which $I(\mathbf{y}; \mathbf{h}_1) \approx I(\mathbf{y}; \mathbf{h}_2) \approx I(\mathbf{y}; \mathbf{h}_3)$, where $I(\mathbf{y}; \mathbf{h}_j)$ is maximised in all layers subject to model constraints. The convergence of all layers to a similar information content shows that *neural networks are good at passing target*

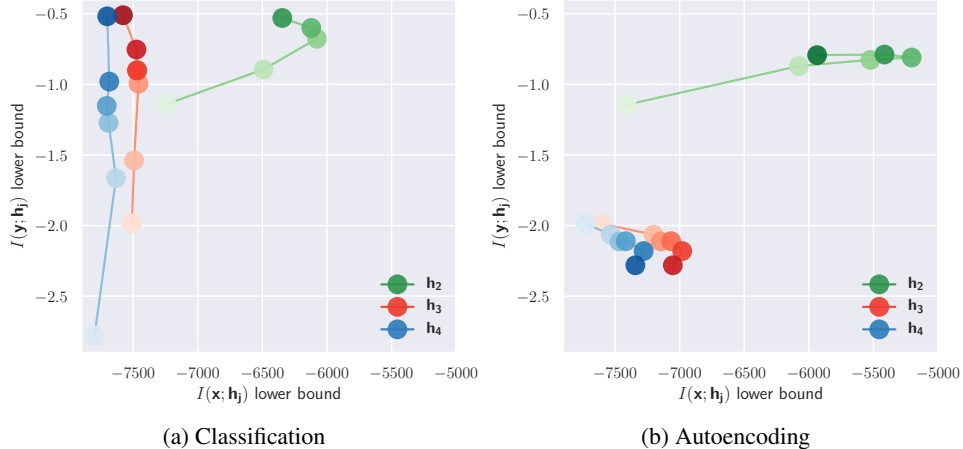

(a) Classification                    (b) Autoencoding

Figure 2: Information curves for comparison with Shwartz-Ziv & Tishby (2017) and Saxe et al. (2018). Classification (a) yields curves like earlier work, while autoencoding curves for $\mathbf{h}_3$ and $\mathbf{h}_4$ do not show the same dynamics. The faint to dark colours represent early to late stage training.

*information forward.* The lighter crosses in Figure 3 (a) are from linear probes (Alain & Bengio, 2016) to show that all layers become more linearly separable while maximising $I(\mathbf{y}; \mathbf{h}_j)$.

*A fitting stage is clear* for all measured layers, where $I(\mathbf{h}; \mathbf{y})$ first increased. This stage was not as short as initially suggested (Shwartz-Ziv & Tishby, 2017) as it took between 5 and 10 epochs. This indicates that the initial fitting phase of learning may have a larger influence on the solution the network finds. The initial representation learning can be seen as learning a good representation that enables compression. For convolutional ResNets, this process is non-trivial.

*We observed compression of information in hidden layers for classification*, shown in Figure 3 (c) by the fact that the MI between input and hidden activations **decreases**. These observations do not necessarily contradict the findings of Saxe et al. (2018), but it does serve to show that compression does indeed occur in this setting. $\mathbf{h}_4$ begins compressing first but also compresses the least (67 $nats$). The layer immediately preceding the average pooling – $\mathbf{h}_3$ – begins compressing between 5 and 10 epochs and compresses almost twice as much (120 $nats$). Finally, the earliest layer we tracked – $\mathbf{h}_2$ – compressed from approximately 10 epochs and to a greater degree than other layers (267 $nats$). Next, we turn our attention to autoencoding since earlier work has focused solely on classification.

**Autoencoding**    We observed compression in hidden layers for autoencoding. Moreover, *class-relevant information* in the bottleneck layer is also compressed (exemplified by $I(\mathbf{y}; \mathbf{h}_3)$). This is because for autoencoding target is class-indifferent. This may explain why simple autoencoding often does not perform well as a pretraining regime for classification *without explicit target-based fine-tuning* (Erhan et al., 2009).

Compression during autoencoding is surprising since the target is identical to the input: there is no target-irrelevant information. An explanation could be that the autoencoder learns a representation that is *easier for decoding*, even at the cost of reducing the MI at the bottleneck layer.

## 6.2    CONDITIONALLY GENERATED PIXELCNN++ SAMPLES

In this section we show and discuss conditionally generated samples to illustrate *the type of information* kept and discarded by a ResNet classifier. Conditional PixelCNN++ Samples were processed for the classifier set-up only, since the samples for the autoencoder were almost entirely indistinguishable from the input. Samples are given in Figures 4 and 5, conditioned on $\mathbf{h}_3$ and $\mathbf{h}_4$, respectively. Samples conditioned on $\mathbf{h}_2$ are given in Appendix D. Inspecting these samples for classifier training gives an intuitive perspective of what quality of information is compressed by the ResNet, in order to classify well. The capacity of $\mathbf{h}_4$ is $16\times$ smaller than $\mathbf{h}_3$ and the samples evidence this. These results serve two purposes: (1) they confirm claims of the IB principle regarding irrelevant infor-

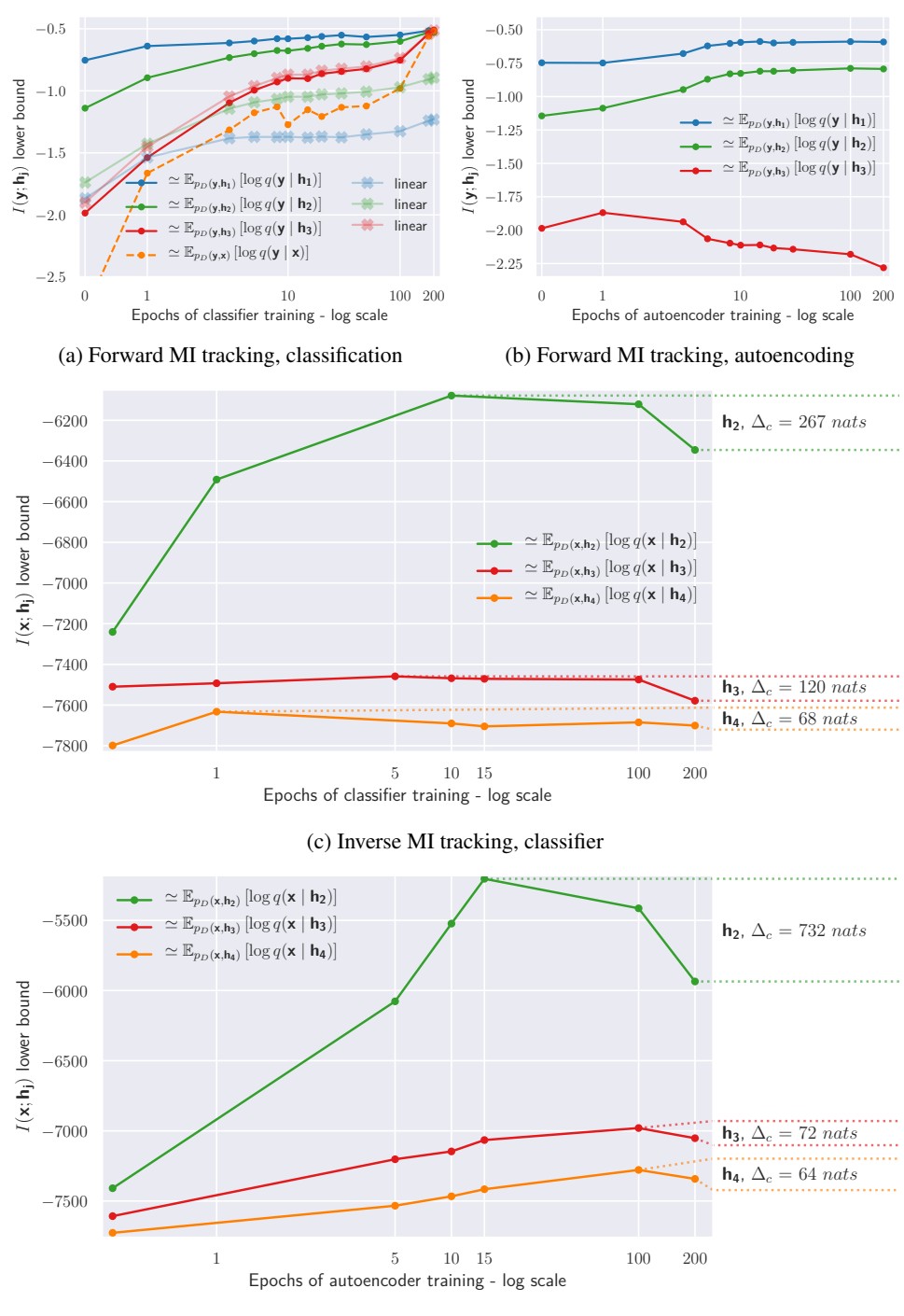

Figure 3: Mutual Information tracking in the forward direction – $I(\mathbf{y}; \mathbf{h}_j)$, (a) and (b) – and inverse direction – $I(\mathbf{x}; \mathbf{h}_j)$, (c) and (d). The classifier always increases MI with the target data (a), while the autoencoder's bottleneck layer compresses label-information. The orange curve in (a) is computed from the classifier's log-likelihood throughout training. Information compression is evidenced in both training regimes, the degree to which is listed as $\Delta_c$ in $nats$ on the right of (c) and (d). Increasing linear separability is also shown in (a). For forward decoding of the autoencoder (b), $I(\mathbf{y}; \mathbf{h}_4) = I(\mathbf{y}; \mathbf{h}_3)$ since the difference in decoding model is only an average pooling operation, which is applied during *encoding* for $\mathbf{h}_4$ and *decoding* for $\mathbf{h}_3$.

mation compression; and (2) they give insight into what image invariances a ResNet learns, which could motivate future work in designing models that target these properties.

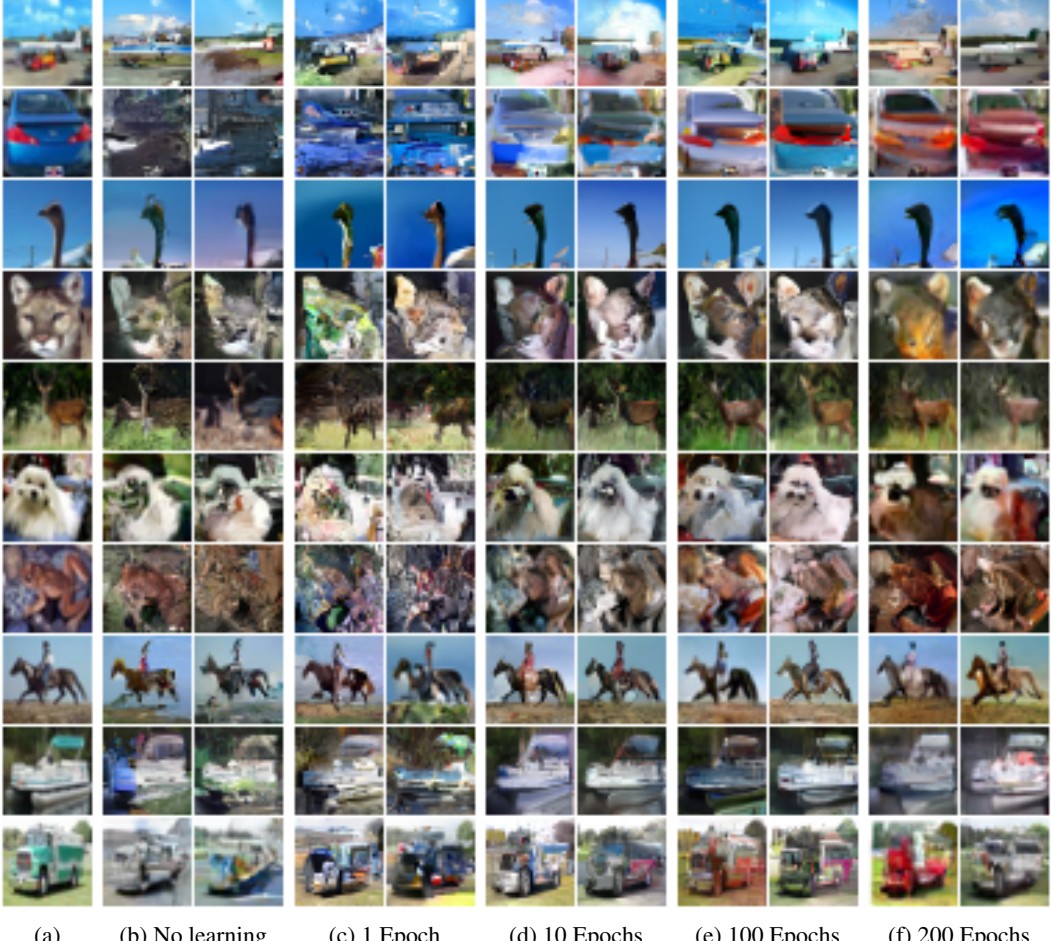

|     |     |     |     |     |     |
| --- | --- | --- | --- | --- | --- |
| (a) | (b) No learning | (c) 1 Epoch | (d) 10 Epochs | (e) 100 Epochs | (f) 200 Epochs |

Figure 4: Samples generated using PixelCNN++, conditioned on $\mathbf{h}_3$ in the **classifier** training set-up. The **original images** processed for $\mathbf{h}_3$ are shown in (a). Ten epochs is close to the peak of the fitting stage, while 200 epochs is the end of learning. Unnecessary features (e.g., background colour) are preserved at 10 epochs, and the sample diversity is greater at 200 epochs. $I(\mathbf{h}_3; \mathbf{x})$ is lower at 200 epochs compared to no learning (Figure 3), but the *quality* of preserved information is better.

**What the network keeps and discards**  Consider that the network compresses information in $\mathbf{h}_3$ such that at its final state there is less information than at initialisation – Figure 3 (c). When inspecting the samples of Figure 4 (b) and (f), we see that even though the information content is higher at network initialisation, the sampled images look like poor renditions of their classes. *The network learns to keep the useful information.* In contrast to this observation we note that *not all irrelevant information is discarded.* The trees behind the truck in the final row of Figure 5 (f) illustrate this.

At initialisation the network weights are random. Even with this random network, information is forward propagated to $\mathbf{h}_4$ enough to generate the samples in Figure 5 (b). Even though these samples share characteristics (such as background colours) with the input, they are not readily recognisable and the shape of the class-relevant components is often lost.

**Colour specific invariances**  Foreground and background *colour partial-invariance* is illustrated in both $\mathbf{h}_3$ and $\mathbf{h}_4$ at the end of training. Consider the car and truck samples to see how the foreground colour of these vehicles is not kept by the layers. The horse samples show background colour

variation. The samples in the deer and truck classes in Figure 5 (f) are still clearly within class but deviate significantly from the input image (a).

The samples conditioned on $\mathbf{h}_3$ later in training, shown in Figure 4 (e) and (f), are more varied than earlier in training. Class irrelevant information – such as the colours of backgrounds (grass, sky, water, etc.) or the specific colour of cars or trucks – is not kept, resulting in more varied samples that nonetheless resemble the input images.

An unconditional PixelCNN++ was also trained for comparison (see Appendix F for loss curves and unconditionally generated samples).

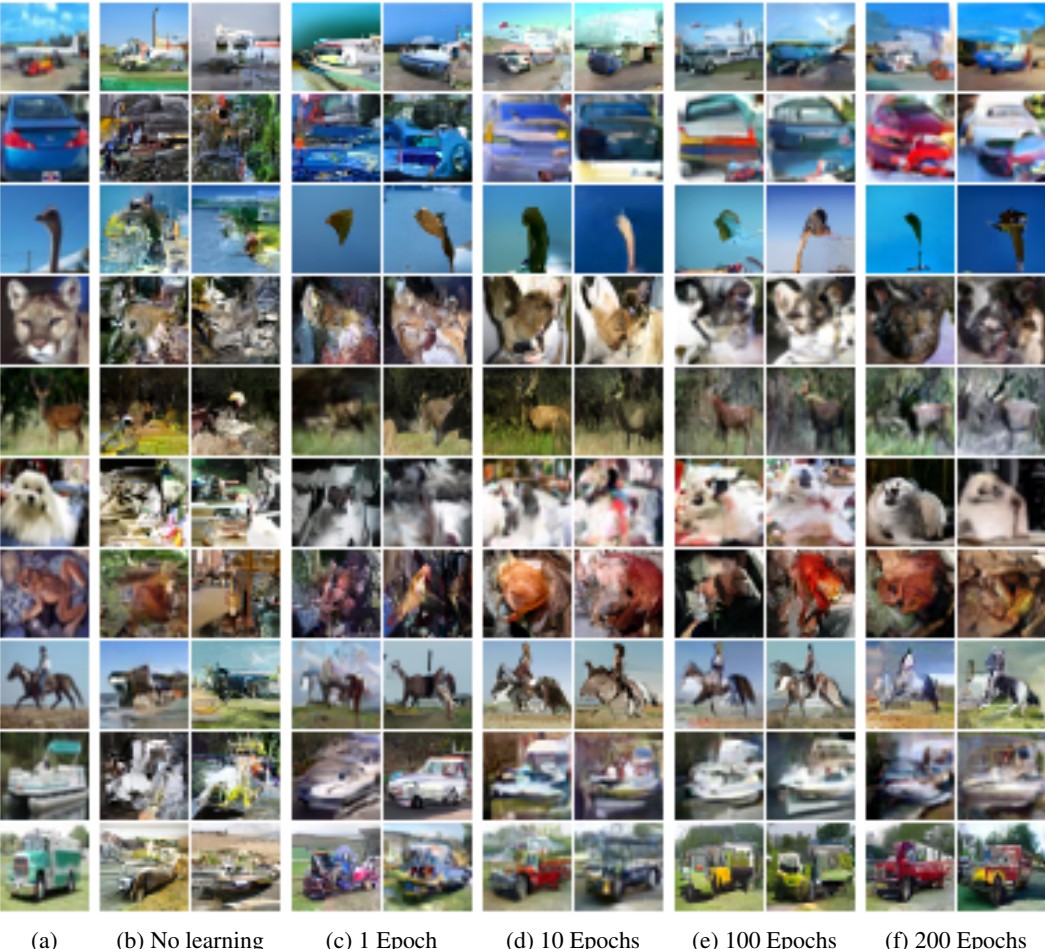

| (a) | (b) No learning | (c) 1 Epoch | (d) 10 Epochs | (e) 100 Epochs | (f) 200 Epochs |

Figure 5: Samples generated using PixelCNN++, conditioned on $\mathbf{h}_4$ in the **classifier** training set-up. The **original images** processed for $\mathbf{h}_3$ are shown in (a).

## 7    DISCUSSION AND CONCLUSION

The ResNet architecture enables very deep CNNs. We show that learning representations using a ResNet results in information compression in hidden layers. We set out in this research to test some of the claims by Shwartz-Ziv & Tishby (2017) regarding the information bottleneck principle applied to deep learning. By defining a lower bound on the MI and 'decoder' models to compute the MI during classifier and autoencoder training regimes, we explored the notion of *compression for generalisation* in the context of realistic images and a modern architecture choice.

For both classification and autoencoding we observed two stages of learning, characterised by: (1) an initial and relatively short-lived increase and (2) a longer decrease in MI between hidden layers

and input training data. Although we cannot confirm the mechanism responsible for compression (*stochastic relaxation*, for example), we gave an intuitive glimpse into what quality/type of information is kept and discarded as ResNets learn. PixelCNN++ models were used to estimate the MI between hidden layers (of the models under scrutiny) and input data; images were generated conditioned on hidden layers to illustrate the fitting and compression of data in a visual and intuitive fashion.

The experimental procedure we developed for this research enables visualising class invariances throughout training. In particular, we see that when a ResNet is maximally (subject to model constraints) compressing information in its hidden layers, the class-irrelevant features of the input images are discarded: conditionally generated samples vary more while retaining information relevant to classification. This result has been shown in theory and for toy examples, but never illustrated to the degree that we do here.

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

## A  ARCHITECTURE

The encoder architecture used in this research is shown in Figure 6.

## B  PIXELCNN++ DECODER MODEL

The original PixelCNN formulation (van den Oord et al., 2016) is autoregressive in the sense that it models an image by decomposing the joint distribution as a product of conditionals:

$$p(\mathbf{x} \mid \mathbf{h}) = \prod_{m=1}^{M^2} p(x_m|\mathbf{h}, x_1, \ldots, x_{m-1}),$$  (4)

where each $x_m$ is a pixel in the image and $\mathbf{h}$ is included for completeness. PixelCNN estimates each colour channel of each pixel using a 255-way softmax function, while the PixelCNN++ improvement does this by estimating a *colour-space* defined by a $K$-way (with $K = 10$ in the original usage) discretized mixture of logistic sigmoids:

$$p(x_m|\boldsymbol{\pi}_m, \boldsymbol{\mu}_m, \mathbb{B}_m) = \sum_{k=1}^{K} \pi_{m_k} \left[ \sigma\left( \frac{x_m + 0.5 - \mu_{m_k}}{s_{m_k}} \right) - \sigma\left( \frac{x_m - 0.5 - \mu_{m_k}}{s_{m_k}} \right) \right],$$  (5)

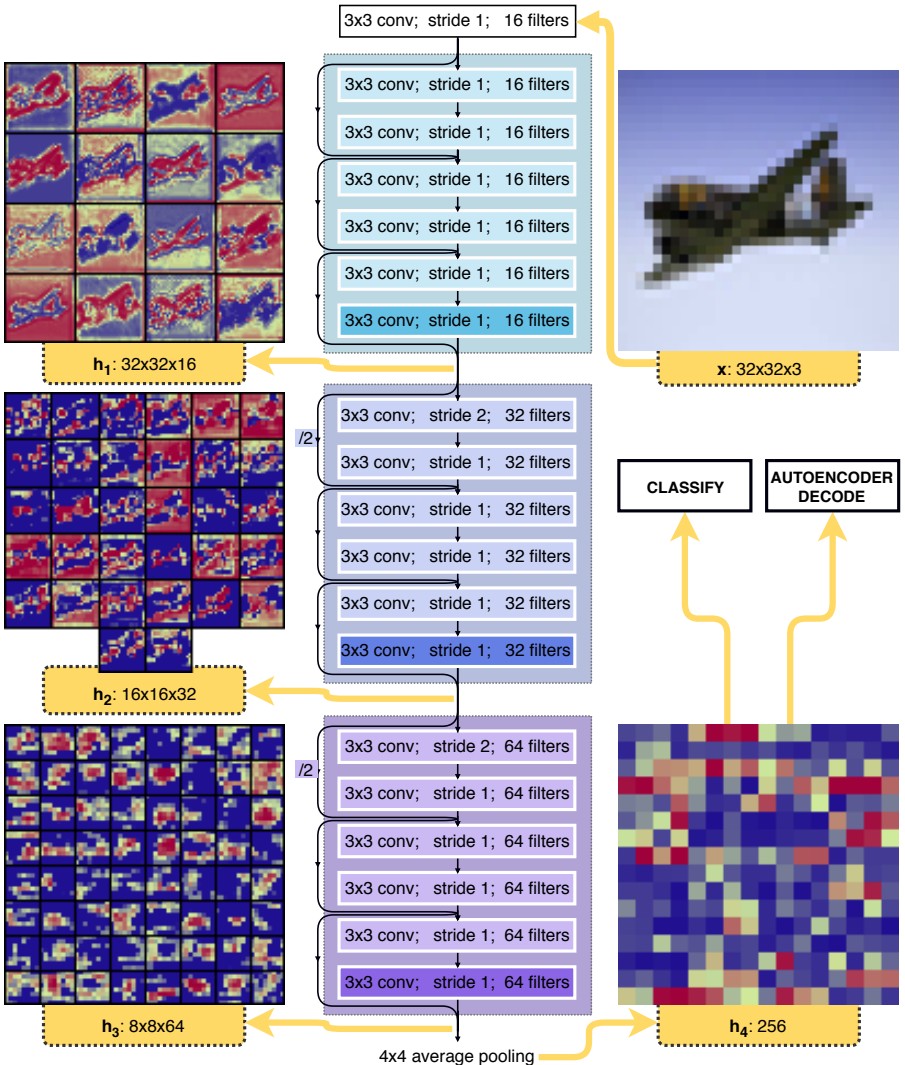

Figure 6: The ResNet model architecture (encoder) used to generate hidden activations in *either* a classification or autoencoder set-up. Each convolution block (inner central blocks) consists of: *convolution → BatchNorm → leaky ReLU non-linearity.* Additional *convolution → BatchNorm* blocks are used at necessary skip connections ('/2' blocks). The hidden representations ($h_1, \ldots, h_4$) are taken at the ends of 'hyper layers', which are the three grouped and separately coloured series of blocks. This encoder is a 21 layer ResNet architecture, accounting for the convolutions in both skip connections.

where $\pi_{m_k}$ is the $k^{\text{th}}$ logistic sigmoid mixture coefficient for pixel $i$, $\mu_{m_k}$ and $s_{m_k}$ are the corresponding mean and scale of the sigmoid, $\sigma(\cdot)$. The discretization is accomplished by binning the network's output within $\pm 0.5$.

The colour channels are coupled by a simple factorisation into three components (red, green, and blue). First, the red channel is predicted using Equation 5. Next, the green channel is predicted in the same way, but the means of the mixture components, $\boldsymbol{\mu}_m$, are allowed to depend on the value of the red pixel. The blue channel depends on both red and green channels in this way.

Salimans et al. (2017) argued that assuming a latent continuous colour intensity and modelling it with a simple continuous distribution (Equation 5) results in more condensed gradient propagation, and a memory efficient predictive distribution for $\mathbf{x}$. Other improvements included down-sampling to capture non-local dependencies and additional skip connections to speed up training.

### B.1 CONDITIONING

The conditioning variable is added to each gated convolution residual block, of which there are six per each of the five hyper-layers in PixelCNN++.

The gated convolution residual block structure was shown empirically to improve results. The activations of a gated residual block are split into two equal parts and one half is processed through a sigmoid function to produce a mask of values in the range $[0, 1]$. This is element-wise multiplied with the other half of the activations as the 'gate'.

As for the conditioning variable, it is conventionally a one-hot encoded class label vector that is added to the activations of each gated residual block. Considering the layers chosen for scrutiny in this work (Figure 6), most of the conditioning variables are three-dimensional: two spatial dimensions and a channel dimension. Additionally, we must account for the down-sampling used in PixelCNN++. Therefore, there are four possible transformations of $\mathbf{h}$ before it can be integrated into the PixelCNN++ model:

1. The **conditioning variable is larger** (regarding spatial width and height) than the activations to which it needs to be added. The conditioning variable is down-sampled using a strided convolution of two and (if necessary) average pooling with a kernel size of two. The filter width is matched in this same convolution.

2. The **conditioning variable is smaller** than the activations. A sub-pixel shuffle convolution (Shi et al., 2016a;b) is used for up-sampling. The sub-pixel shuffle is an alternative to deconvolution or nearest neighbour up-sampling that allows the model to learn the correct up-sampling without unnecessary padding. A non-strided convolution with a kernel of one matches the filter width.

3. The **conditioning variable is the same size** as the activations. A non-strided convolution with a kernel of one matches the filter width.

4. The **conditioning variable is, instead, a vector** – $\mathbf{h}_4$ in Figure 6. The dot product of these and the appropriately sized weight matrix are taken to match the activations.

If, for example, $\mathbf{h} = \mathbf{h}_2$ is a $(16 \times 16) \times 32$ (two-dimensional with 32 filters, the second hyper-layer in Figure 6) hidden representation, the first three aforementioned transformations would be in effect because the configuration of PixelCNN++ (Salimans et al., 2017) means that there are activations with spatial resolutions of $(32 \times 32)$, $(16 \times 16)$, and $(8 \times 8)$, to which the conditioning variable must be added.

## C    SUMMARY OF EXPERIMENTAL APPROACH

## D    MORE SAMPLES

Figure 7 shows conditional samples generated by conditioning PixelCNN++ models on $\mathbf{h}_2$ (see Figure 6), respectively. $\mathbf{h}_2$ showed the most compression (Figure 3) but the quality of information that was compressed clearly did not influence the structure of the samples. Instead, global hue and

---

**Algorithm 1** Experimental procedure.

1: **procedure** TRACK MI($model = classifier$)
2:     **for** $epoch$ **from** 1 **to** 200 **do**
3:         $modelParameters \leftarrow$ train($model, D_{encoding}$)
4:         **if** require MI query **then**
5:             **for** $\mathbf{h}_j$ **in** $\mathbf{h}_{all}$ **do**
6:                 freeze($model, layer = \mathbf{h}_j$)
7:                 train($decoderForward, D_{decoding}$)          ▷ partially frozen classifer
8:                 train($decoderInverse, D_{decoding}$)              ▷ PixelCNN++
9:                 $MI^{forward}_{\mathbf{h}_j, epoch} \leftarrow$ estimateMI($decoderForward, D_{evaluation}$)
10:                $MI^{inverse}_{\mathbf{h}_j, epoch} \leftarrow$ estimateMI($decoderInverse, D_{evaluation}$)

---

colour variations were influenced at this layer. This comparison is important because it evidences that merely measuring the MI alone does not give an encompassing perspective of what type and quality of information is compressed in a CNN.

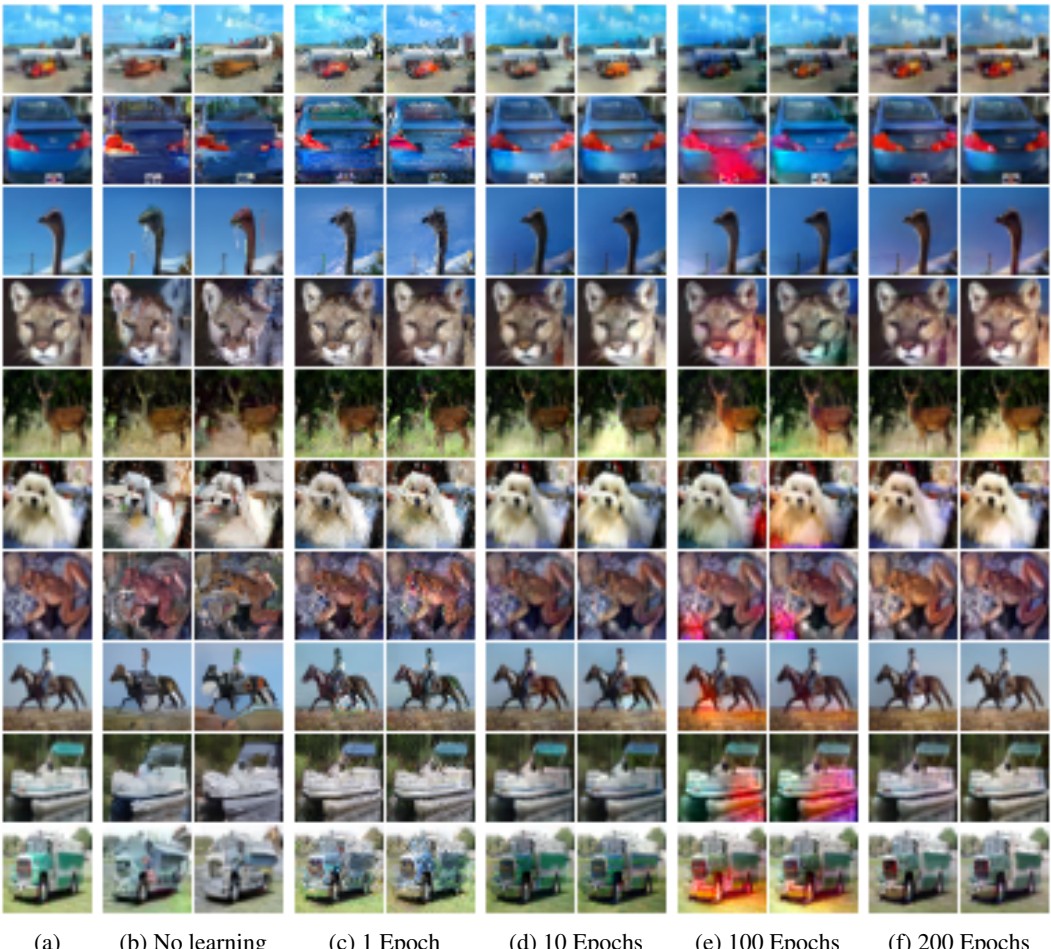

    (a)       (b) No learning     (c) 1 Epoch     (d) 10 Epochs    (e) 100 Epochs    (f) 200 Epochs

Figure 7: Samples generated using PixelCNN++, conditioned on $\mathbf{h}_2$ in the **classifier** training set-up. The **original images** processed for $\mathbf{h}_3$ are shown in (a).

# E PixelCNN++ Training Curves

Figures 8 and 9 exemplify the convergence regime for the PixelCNN++ decoding models. Even after 250 epochs of training convergence was yet to be reached. On a single Titan 1080ti GPU each of these training runs took approximately 15 days.

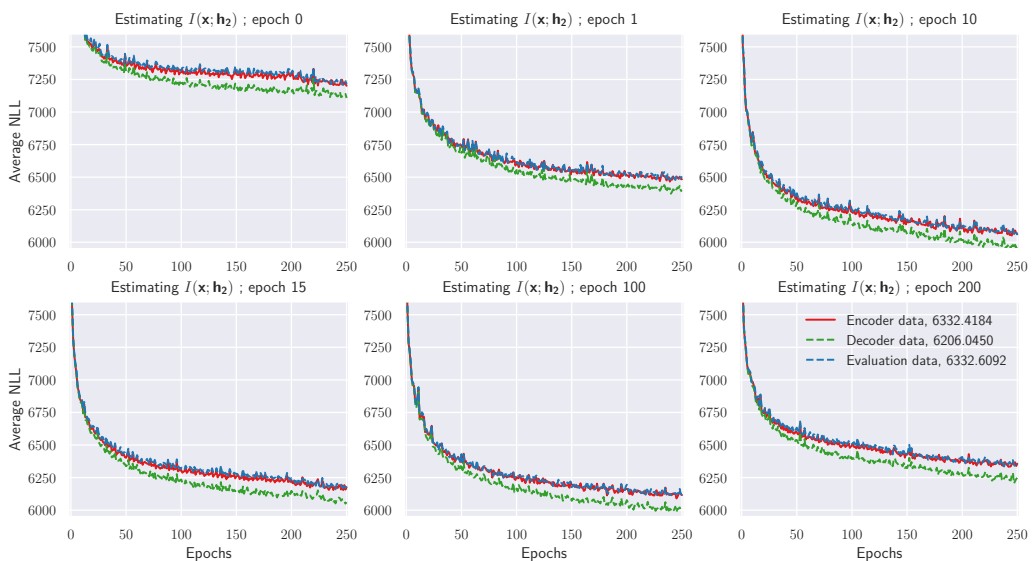

Figure 8: PixelCNN++ decoder models' loss curves for estimating $I(\mathbf{x}; \mathbf{h}_2)$, for classifier training. Each set of curves shows the decoding run for one data point in Figure 3. These models were stopped at 250 epochs of decoding owing to time and computation constraints.

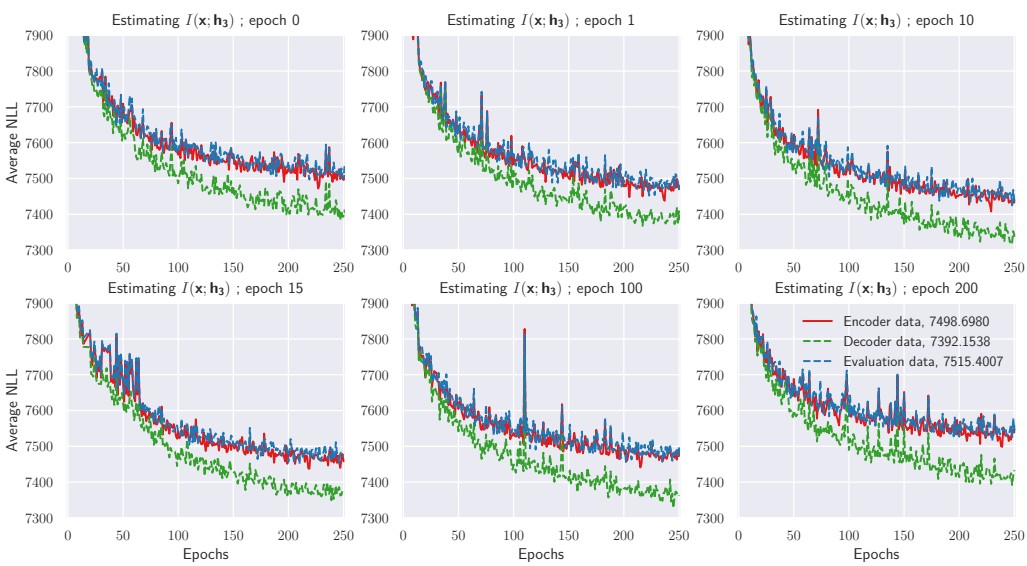

Figure 9: PixelCNN++ decoder models' loss curves for estimating $I(\mathbf{x}; \mathbf{h}_3)$, for classifier training. Each set of curves shows the decoding run for one data point in Figure 3. These models were stopped at 250 epochs of decoding owing to time and computation constraints.

## F UNCONDITIONAL PIXELCNN++

Figure 10 shows the training curves for an unconditional PixelCNN++ trained on the encoder dataset of CINIC-10. Samples generated are shown in Figure 11, giving context and scale to the type of samples generated in this work.

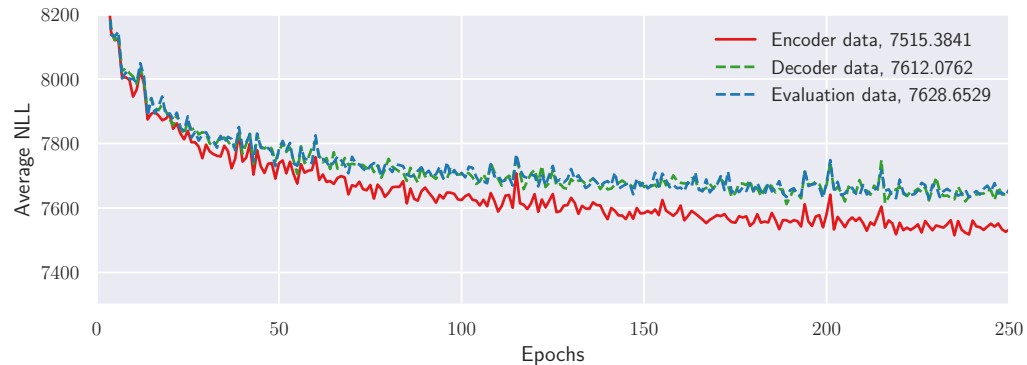

Figure 10: Unconditional PixelCNN++ loss curves when trained on the encoder dataset of CINIC-10. Since this is only using one third of CINIC-10, it may be possible to achieve a lower loss when using a larger portion of CINIC-10. The best evaluation loss here corresponds to 3.58 bits per dimension, as opposed to the 2.92 bits per dimension on CIFAR-10 (Salimans et al., 2017).

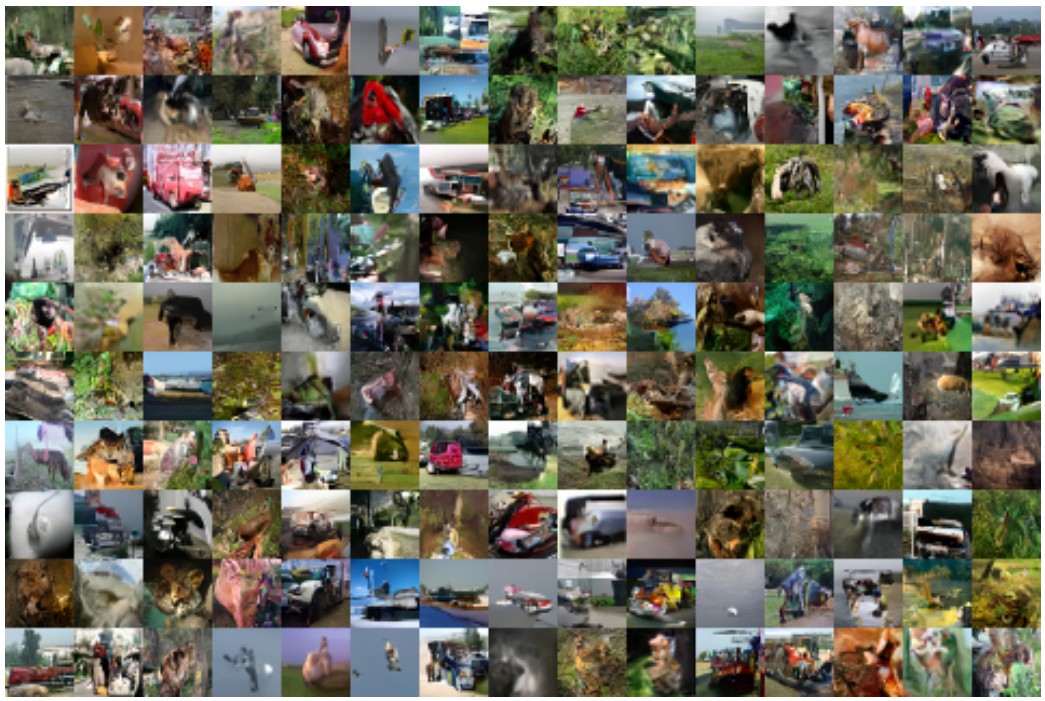

Figure 11: Unconditional PixelCNN++ generated samples when trained on the encoder dataset of CINIC-10. These samples have good local qualities but are not particularly convincing as real images. This is a known pitfall of autoregressive explicit density estimators.

