# OpenReview forum: "What Information Does a ResNet Compress?"
_ICLR.cc/2019/Conference_

### Official Review · AnonReviewer3 · 2018-11-02
**Interesting empirical work on compression in Resnets with partially inconclusive results**

**Rating:** 6
**Confidence:** 5

**Review:**

-> Summary

The authors propose to extend the analysis of Shwartz-Ziv & Tishby on the information bottleneck principle in artificial neural network training to realistic large-scale settings. They do so by replacing otherwise intractable quantities with tractable bounds in forms of classifiers for I(y;h) and Pixel CNNs for I(x;h). In conclusion, they observe two phases during training, one that maximizes mutual information between input and hidden representation and a second one that compresses the representation at the end of training, in line with the predictions from toy tasks of Shwartz-Ziv & Tishby.

-> Quality

The paper is very well written, all concepts are well-motivated and explained.

-> Significance

The main novelty is to replace intractable quantities in the analysis of the information bottleneck with tractable bounds in form of auxiliary models. The idea is neat and makes a lot of sense. On the other hand, some of the results and the bounds themselves are well-known and can thus not be considered novel. The main contribution is thus the empirical analysis itself and given some overly confident claims on qualitative results and missing ablation on the quantitative side, I am not convinced that the overall results are very conclusive.

-> Main Concerns

The authors make a lot of claims about the qualitative diversity of samples from deeper layers h4 of the network as compared to h1 and h2. However, I do not agree with this. When I look at the samples I see a lot of variations early in training and also in layers h1 and h2. The difference to h4 seems marginal at best and not as clear cut as the authors present it. Thus, these claims should be softened.

In figure 1 I tend to say that samples at epoch 1 are more varied than at epoch 200. In figure 5 (b) seems pretty color invariant and not only (f) as claimed. In fact (f) seems pretty stable and consistent to me.

The bound in equation (2) might be quite loose, depending on the quality of the classifier or pixel CNN. Even though there is no way to test this, it should be discussed.

What is the effect of weight decay here? I suspect that weight decay plays a crucial role in the final compression phase observed in e.g. figure 3 (c), but might not be a necessary condition to make the network generalize. An ablation experiment verifying or falsifying this statement would be important to conduct and without it I am not convinced that the shown curves are conclusive.

-> Minor

- You seem to use a weird math font, is this on purpose? It does not seem to be the ICLR standard.
- The bound in equation (2) is a standard variational bound and has been used many times, the authors make it sound like it is their contribution. You should maybe cite basic work and recent work on variational information bottleneck here.

---

> ### Public Comment · (anonymous) · 2018-11-19
> **Response to AnonReviewer3**
>
> Thank you for your review. Minor comments have been accounted for in the amended script. This will be uploaded as a revision shortly. Major comments will be dealt with here.
>
> Regarding missing ablation studies. It takes approximately two weeks to train a single PixelCNN++ model for a single data point in Figures 3c and d. We intentionally chose a PixelCNN++ model and the CINIC-10 dataset to ensure the analysis we undertook was as thorough as possible, and that the bounds on the MI were as tight as possible, given current state of the art research and computational feasibilities for this estimation. Ablations studies are simply computationally infeasible for us.
>
> Regarding the comments on the generated samples. It seems that what is missing from our discussion is the caveat that at convergence the images most resemble the classes they are in. Take Figure 1 for example: you are likely correct that there is more variation at epoch 1 than at epoch 200, but that variation is largely noise related and not necessarily variation in the specific characteristics that would be more natural. At 200 epochs in this figure, consider that the background (both the ground and the sky) are smooth and arguably more realistic looking than at 1 epoch, yet they still vary from sample to sample. A similar argument can be made for the horse itself, in terms of both colour and somewhat in head shape. In the paper itself we said “When inspecting the samples of Figure 4 (b) and (f), we see that even though the information content is higher at network initialisation, the sampled images look like poor renditions of their classes.”
>
> We understand that this is a rather subjective perspective on some generated samples, but we are trying to provide insight based on what is available in this analysis. Attempting to understand information compression in a ResNet on these images is a challenging task, hence we value your perspective and input here.
>
> Regarding the effect of weight decay. We will have to keep this for future work owing to computational constraints. That said, our stance was when where we adopted a modern, well-known, and widely used architecture and training scheme and analysed it as it stood. There are many changes that could be made - removing batch norm, varying the initialisation scheme, weight decay, optimiser preferences, etc. - but these create a scenario where the number of experiments to run becomes combinatorial. Since each PixelCNN++ model takes two weeks to train on a Titan 1080 GPU, compromises were made.

---

> > ### Comment · AnonReviewer3 · 2018-11-20
> > **Thank you for your detailed answer**
> >
> > You seem to agree that one of your major claims is not as clear-cut as your wording in the manuscript suggests and that the interpretation of the samples is rather subjective. However, you still make relatively large claims about them, e.g.:
> >
> > "Sampling images by conditioning on hidden layers’ activations offers an intuitive visualization to understand what a ResNets learns to forget."
> >
> > "Analysis of PixelCNN++ samples conditioned on hidden layer activations to illustrate the type of information that ResNet classifiers learn to compress. This is done via the visual demonstration of the sorts of invariances that a ResNet learns."
> >
> > "...they give insight into what image invariances a ResNet learns..."
> >
> > etc.
> >
> > These claims should be softened and it should be stated, especially in the abstract, that the conclusions drawn here are rather subjective or difficult to interpret, at least the way they are presented right now.
> >
> > --------
> > minor:
> > - x and y in equation (1) still look strange

---

> > > ### Author Response · Authors · 2018-11-21
> > > **Claims**
> > >
> > > Thank you for your response.
> > >
> > > The specific sentences you quote there still hold: this manner of visually inspecting a modern network's processing (a ResNet, here) does offer a unique and intuitive insight into the learning process. Those sentences suggest what can be and is done through using this information theoretic approach to analysis, but leave much of the analysis to the skills and interpretation of the reader. That is the nature of such a visualisation.
> > >
> > > Regarding the math font used: we use serif font for random variables/vectors to distinguish quantities such as h (hidden representation) and H (information entropy). This is not an uncommon font usage and we have been consistent throughout the script.

---

> > > > ### Comment · AnonReviewer3 · 2018-11-21
> > > > **Thank you for the discussion**
> > > >
> > > > It is hard for me to draw a (subjective) conclusion from the qualitative samples, which is a significant downside of the submission. I wish the authors would have come up with a better experimental setup for this to allow clearer statements, I do realize that this might be hard to do though.
> > > > On the other hand, I do find the quantitative analysis based on the Pixel-CNN likelihoods valuable and the observed compression as measured by the generative model is interesting and seems relatively consistent. However, a significant downside is the infeasibility of the method when it comes to more fine-grained ablation studies.
> > > >
> > > > All in all, I will keep my rating as is, because it reflects the pros and cons I see in this paper.

---

### Official Review · AnonReviewer1 · 2018-11-03
**An empirical study with specious results**

**Rating:** 4
**Confidence:** 3

**Review:**

## Summary

This paper is an empirical study which attempts to test some of the claims regarding the information bottleneck principle applied to deep learning. To estimate the mutual information (I(x; h) and I(y; h)) in neural networks, the authors define a lower bound on the MI. Then a PixelCNN++ model (for I(x; h)) and a partially frozen classifier (for I(y; h)) are used to compute the MI during classifier and autoencoder training. For both tasks, the authors report the mutual information between hidden layers and input training data first increase for a while and then decrease. The generated images conditioned on hidden layers by the PixelCNN++ were shown to demonstrate the fitting and compression of data in a visual and intuitive fashion.

In general, the paper is well-written and organized. The idea behind the paper is not novel. Shwartz-Ziv & Tishby (2017) and Nash et al. (2018) also attempt to test the information bottleneck principle by estimating the mutual information. The results of this paper are specious and hard to be explained.

## Issues with the tightness of the lower bound
The tightness of the lower bound is dependent on the KL divergence between the true conditional distribution p(x|h) and the approximating distribution q(x|h). Does the adopted PixelCNN++ is good enough to approximate the true conditional distribution? There is not any discussion.

## Issues with the results of autoencoder
The decrease of the mutual information in autoencoder training is very specious. Since the decoder part of autoencoder should generate better and better images during the training process, does it mean that the PixelCNN++ was worse? Does it imply that the optimization of the PixelCNN++ has some unknown problems?

## Issues with the connection between this paper and Nash et al. (2018)
These two paper have used the same lower bound and the same PixelCNN++ for estimating the mutual information. The observations are also similar. Both of these papers found the mutual information between inputs and network layers decreases over the training. The differences of these two papers are the adopted neural networks and the dataset, which are kind of minor.

---

> ### Public Comment · (anonymous) · 2018-11-19
> **Response to AnonReviewer1**
>
> Thank you for your review. Amendments and alterations have been made to the paper to account for minor changes. This will be uploaded as a revision shortly. Regarding the major points of the review, we will address these here.
>
> It is unclear from this review what about the results are specious and hard to explain.
>
> Regarding the tightness of the bound. There is no way of theoretically knowing how good the bound is. Another perspective to take is that of USABLE information and the analysis thereof, which is arguably in line with the perspective of Schwartz-Ziv & Tishby. That is, just because information is present, does not mean it is useful or even accessible. As an example of this, consider the process of hashing a password: the information is retained and recoverable but useless without the correct access. A PixelCNN++ is state of the art at extracting usable information as an explicit image distribution estimator. It is unlikely that any other recent models would do better. Therefore, we chose to trade computation for as tight of a bound as we could. In fact, it takes approximately two weeks to train a single PixelCNN++ model (which yields a single data point in Figure 3c and d) to interpret the mutual information here.
>
> Regarding the autoencoder. We agree that these results can seem counter-intuitive. However, there are a number of possible explanations for these:
> 1. The autoencoder is learning a representation at the bottleneck for which it is easier for a decoder to learn even at the cost of reducing MI (we mentioned this in the paper already).
> 2. The mean-squared-error loss criterion is not well suited to preserving information. This issue is similar to the ‘mode averaging’ problem that causes blurry reconstructions in an autoencoder trained with this loss. Essentially, the target and the reconstruction are different owing to an imperfect loss function and therefore information is discarded.
> 3. Some global components are easier to reconstruct than local features. This is nearly the same as the previous point, but since the MI computation accounts for every pixel in the reconstruction, unless sharp details are being modeled by the autoencoder (which they’re not), information will be lost.
>
> Regarding issues with the connection of this paper with Nash et. al: these were undertaken concurrently and should be consider as such.

---

> > ### Comment · AnonReviewer1 · 2018-11-29
> > **Thank you for the response**
> >
> > Using a powerful model (like PixelCNN++) is not solving the tightness issue. Since the results of autoencoder experiments could be explained in different ways and have inherent flaws (for example the loss), I suggest to remove this part from this work.
> >
> > After reading the response, I will keep my rating.

---

### Official Review · AnonReviewer2 · 2018-11-03
**Empirical evaluation of information retained across layers of classification ResNets using pixelCNN decoders.**

**Rating:** 4
**Confidence:** 4

**Review:**


* Summary:

This work is an empirical study of the relevance of the Information Bottleneck principle as a way of understanding deep-learning. It is carried out in the setting of realistically sized networks trained on natural images dataset. This is, in spirit, a meaningful and sensible contribution to the ongoing debate. Being a largely empirical contribution, its value hinges on the exhaustivity and clarity of the experiments carried out. As it stands, I believe that these should be, and can be, improved. Details to support this opinion are given below. A summary of my expectations is given at the end.


* Summary of the approach and significance:

The IB principle relies on estimating the mutual information between i) the input and an intermediate layer, I(x,h), and an intermediate layer and the output, I(h, y). Previous work has relied on binning strategies to estimate these quantities. This is not applicable in a real-sized problem such as classification of natural images with deep networks. This paper proposes to invert a first deep model using a second, generative, model which must reconstruct the input of the first given some intermediate layer. The information progressively discarded by the first network should be modelled as uncertainty by the second. This yields a lower bound on the mutual information, with a tightness that depends on the expressivity of the generative model.

I believe the goal to be meaningful and a valuable contribution: going forward, testing this assumption in realistic setting is essential to the debate. The proposed approach to do this seems sensible to me. It is similar to cited work by Nash et al., however both works are concurrent and so far unpublished and should be considered as complementary point of views on the same problem.

Partial conclusion: The goal is meaningful and sensible.

* Quality of writing.

In sections 1, 2 and 3, the motivation is clear and contains relevant information. I feel it could be polished further. In particular, some redundant information could be condensed. The introduction to the IB principle, though understandable, could be improved. Here are some opinions:

>> Paragraphs 3 and 4 of 1.0: A reference to M. Saxe et al. could already be made there: it is a major 'opponent' in the debate to which you are empirically contributing.

>> In paragraph 1.1: Points 1) and 2) are redundant. So are 3) and 4). Paragraph 1.1 as a whole is largely redundant with the previous paragraph, these could be collapsed.

>> In section 2: I feel that the main intuitions of the encoder / decoder distributions (I(y,h) / I(x,h)), the information plane, and the optimal bottleneck representations (Sections 2.1 to 2.3 of Schwartz-Ziv & Tishby) could be better conveyed in 2, though I understand the need for brevity.

>> Related works: Descriptions of related works could be condensed. On the other hands, the points of these papers that you contradict in you experiment section could be explicitly mentioned again there.

>> End of section 3, section 4: The fact that estimation of the MI by traditional means is not applicable in your setting is repeated many times throughout the paper, in noticeably rapid succession in that region. Some mentions should be removed.

### More minor points:

>> Paragraphs 2 of 1.0:
- 'the extraction of typical abstract properties...' this statement is vague and thus debatable: the channel-wise mean in RGB space, for instance, is not an especially abstract property.
- Reference is made to Zhang et al. to justify the need for more analysis of generalization in deep-learning. This paper can be considered controversial. Mention could be made of other works, for instance about the inaplicability of traditional generalization bounds and attempts to improve them.

>> Pseudo code algorithm.
- Could be summarized in the main body and pushed to the annex.

>> The choice of pixCNN as a generative model could be discussed more. There are some good reasons to prefer this to L2 regression for instance.

Partial conclusion: The description of the method contains relevant information and is functional, but the writing could be improved.


* Experimental results.

> The contribution and novelty of this paper is largely empirical. Therefore the experimental results should be held to a high standard of clarity and exhaustivity.


*** The choice of dataset:
The experimental setup seems to be fair in terms of dataset / split chosen: the abundance of data for the three steps (encoding, decoding, evaluation) is a notable strength.

*** The quality of the lower bound: Uncertainty when reconstructing the image may come from the fact that information has been discarded. Variance may also come from the pixCNN++, which is imperfect. You mention this (paragraph 4.1) but do not take experimental steps to measure it. Please consider reporting the performance of your generative model i) without conditioning, ii) conditioned on one-hot ground truth labels, and optionally iii) on grayscale/downsampled versions of the image without otherwise modifying the training setup. These values will give the reader an idea of the significance of variations in MI measured and give a 'scale' to your figures, strengthening your claims.

*** The evolution of compression *accross iterations* for a fixed layer
I will focus on the classification setting for now.

Qualitatively: Figures 1, 4 and 5 do not convince me that a meaningful evolution in the type of information discarded *across training iterations* can be observed visually. In figures 1 and 4, the network seems to learn invariances to some coloring, and its preferred colours vary across iterations. Beyond that I cannot see much, except maybe for column (f) of figure 4, despite your claim in section 2, paragraph 2.

Quantitatively: Curves in Figure 2 a) are more convincing, though a notion of scale is missing, as already discussed. The evolution of I(y; h) across iterations is very clear, in Figure 2 a) and especially 3 a). The evolution of I(x, h) much less so. h3 and h4 do not seem to show anything meaningful. In h2 the decrease in I(x, h) is supported by only 2 points in the curve (epoch 10 to epoch 100, and epoch 100 to epoch 200, figures 2a and 3c). Epochs displayed are also incoherent from one curve to the next (epoch 15 is missing for h2 in fig 3c) which raises suspicion. It appears important to i) display more points, to show that this is not just noise and ii) track more layers to confirm the trend, supported by a single layer so far (see next paragraph). I understand that each of these points require training of a generative model, but I feel it is necessary to make reliable conclusions.

Minor: In figure 2, epochs should be added as labels to the colours.

*** The evolution of compression *across layers* for a fixed iteration

Conversely, the evolution of the MI across layers is very convincingly demonstrated, and I feel this is perhaps the main strength of this paper. All curves display consistent trends across layers, and Figure 5 qualitatively displays much more invariance to pose, detail, etc than Figure 4. This is interesting, and could be made more central: i) by making a figure that compares samples across layers, for a fixed iteration, side by side.

On the downside, I believe it is important to track more layers, as it is to me the main interest of your results. The second paragraph of section 5 does not give a good idea of the spread of these layers to someone not familiar with the resnet architecture used. For example, the penultimate layer of the network could be used (the layer at which the most compression is to be expected).

*** On the auto-encoder experiments.

> Little detail is given about the way the auto-encoder is constructed. In particular, one expects the type of bottleneck used (necessary so that the network does not learn the identity function) to have large impact on the amount of information discarded in the encoding process. This dependency is not discussed. More crucially, experiments with different types / strength of bottleneck are not given, and would, in my opinion, be key to an analysis of this dependency through the IB principle.

> Furthermore, no qualitative analysis is provided in this setting.

> Without these additions, I find the Auto-encoding setting an unconvincing distraction from the main contribution of this paper.

***  main avenues of improvement:

> Two kinds of progression in compression are demonstrated in your paper: across layers, and across iterations.
As it stands, results evidence the former more convincingly than the latter, both qualitatively and quantitatively.
I believe results could be presented in a way that clearly takes better advantage of this, as I will detail further.
More data points (across layer and epochs) would be beneficial. I feel that the auto-encoder setting, as it stands, is a distraction.
I would find this paper more convincing if experiments focused more on showing how layers progressively discard information, and less on the 'training phases' that are so far less clear.

*** Additional comments

The following are a number of points that would be worthwhile to discuss in the paper

> As it stands, it seems the line of reasoning and experimental setup seems to rely on the chain-structured nature of the considered neural net architecture. Can the same line of reasoning be applied to networks with more general computational graphs, such as dense-nets [a], mulit-scale denseness [b], fractal nets [c] etc.

[a] Huang, G.; Liu, Z.; van der Maaten, L. & Weinberger, K. Densely connected convolutional networks CVPR, 2017
[b] Huang, G.; Chen, D.; Li, T.; Wu, F.; van der Maaten, L. & Weinberger, K. Multi-Scale Dense Networks for Resource Efficient Image Classification ICLR, 2018
[c] https://arxiv.org/abs/1605.07648

> Why is it that earlier layers are estimated to have larger MI with the target y than later layers before convergence? Sure, later layers compress certain information about the input x, which could be informative on the response variable y. But since the MI estimate for early layers depends on the same network architecture as the one used to compute the later layers from the early ones, the result seems counter intuitive. See paragraph "forward direction" in section 4.1.

> The orange curve in fig 3a estimating I(x;y) is not commented upon. How was it obtained, what is its relevance to the discussion?

---

> ### Public Comment · (anonymous) · 2018-11-19
> **Response to AnonReviewer2**
>
> Thank you for your review. Amendments and additions have been made throughout the paper to account for the minor points and suggestions. This will be uploaded as a revision shortly. We will now discuss the major points.
>
> First, we would like to make apparent the computational burden of in depth analysis in this scenario: each PixelCNN++ model takes approximately two weeks to train on a single Titan 1080ti GPU. This, for example, is the reason there are slightly more data points for h3 in Figure 3c - we determined that this layer was likely to yield interesting results since it is the penultimate layer of the ResNet.
>
> Regarding the quality of the bound on I(x, h). There is no way of theoretically knowing how good the bound is. However, PixelCNN++ is state of the art for extracting useful information in this scenario.
>
> Regarding the evolution of information across iterations for a fixed layer. First, consider Figure 1:
> - at very early stages of training (epochs 0 and 1) the generated samples are noisy and less recognisable as horses when compared to later stages.
> - compare the samples at 10 epochs (roughly the peak of ‘fitting’) to those at 200 epochs (end of compression). The background (ground and sky) is less varied earlier on; the colour of the horse is less varied earlier on (notwithstanding noise such as the first row of d); and the positioning of the horses head is less varied earlier on.
>
> Unfortunately these changes are difficult to see unless your PDF viewer does not interpolate pixels and you can zoom sufficiently. It would be ideal to run these experiments on higher resolution images, but simply not computationally feasible at present.
> The same sentiments are true for Figure 4 (comparing column (d) and (f), most notably), but owing to the already limited information after average pooling for Figure 5, it is difficult to interpret in the same fashion. Nonetheless, the quality of the images is notably different early and late stage training.
>
> Regarding the suggestion of more focus on the evolution of information across layers for a fixed iteration. This is largely model specific based on the capacity of each layer and an expected outcome given the structure of this ResNet architecture. Taking an approach of focus on this sort of information evolution is not the direction of this paper, particularly since it was written for comparison with Schwartz-Ziv & Tishby. Future work focusing on this suggested evolution can be undertaken.
>
> Regarding the spread of h2, h3, and h4 in the ResNet. This is more clear in Figure 6 in the appendix, but we have made it more obvious in the text, too. Specifically, h4 is the penultimate layer and is that layer which should exhibit the most compression.
>
> Regarding the autoencoder set-up. The decoder of the autoencoder was designed to be as close to an inversion of the encoder structure as possible. Therefore, the bottleneck is defined as the average pooling layer of the ResNet itself. We have made this more clear in the text. Figure 6 in the appendix is an architecture description. Any sort of ablation studies and further hyper-parameter adjustment requires training more PixelCNN++ models for analysis, which is computationally infeasible. Regarding generated images from the h layers for the autoencoder: these almost always look indistinguishable from the input images, so we kept these out for brevity. Finally, since the autoencoder set-up is not comparable to earlier research, we sought to keep these results brief. Nonetheless, the evidence that compression occurs in an autoencoder was an interesting finding and we chose to keep these results in the paper.
>
> Regarding the reason why earlier layers have higher I(y, h) than later layers before convergence. This is because of the data processing inequality. Specifically, information can only be discarded and never gained. Consider early stage training before convergence. None of the layers are doing a particularly good job of retaining information about y, iteratively throwing information away. Earlier layers will see representations that have a lower level of information degradation and, therefore, always have more information than later layers. As the network learns and approaches convergence, each layer becomes better and better at passing y-relevant information forward and less information gets discarded (they reach a very similar I(y, h) point at 200 epochs). Figure 3a is quite indicative of what you would expect layers to do regarding class-relevant information retention over training.
>
> Regarding the orange curve in Figure 3a. This curve is the original training curve of the network under scrutiny and is directly related to the log-likelihood of the model. We will make this clearer in the caption.

---

> > ### Comment · AnonReviewer2 · 2018-11-22
> > **Training complexity and MI curves**
> >
> > Hi, thank you for your answers above.
> > Let me come back to a few points below.
> >
> > I do appreciate the computational burden of getting these results. In my experience, using the code provided by the authors, pixCNN++ reaches 2.98 bpd in roughly 12 hours on a gtx1080ti, then reaches 2.96 in 24 hours, and converges to 2.95 bpd in 5 days. It requires multi-gpu setups (for increased batch size) and 5 days to reach 2.92. Assuming the same tendency is observed with your inputs, training all models with a fixed budget of 24 hours would seem to be enough to conclude. This is still heavy, but an order of magnitude more manageable than what you describe. I stand by my initial opinion that more points are required for the results to be convincing.
> >
> >
> > > Regarding the reason why earlier layers have higher I(y, h) than later layers before convergence. This is because of the data processing inequality [...]
> >
> > The fact that the curves are ordered as predicted by the data processing inequality is very clear. My question regarded the way this estimation is obtained, but was unclear and I will try to reformulate it here.
> > Suppose that one wants to estimate I(y, h_i) and I(y, h_j) with j>i. Given that I(y,h_i) is estimated by freezing the weights for layers {h_k}_{k<=i} and retraining further layers, intuition could suggest
> > that the layers between h_i and h_j will throw exactly the same amount of information when retraining than what they did when training in the first place, such that h_j in the first network and h_j in the
> > network retrained to estimate I(y, h_i) will contain the same amount of information. And indeed, as training progresses, all MI estimates seem to converge to very close values (last point of figure 3a), such that the data
> > processing inequality almost becomes and equality. Could you explain the intuition of why one should expect the retrained network to not have this behaviour?
> > If this reasoning is not flawed, it raises an other question. The fact that all curves converge to the same value could be due to the fact that the network learns to pass all the relevant information along, as you concluded.
> > But it could also be due to the fact that the method used to estimate it throws away the same amount of information as the first network, irrespective of the starting point. I would like this to be clarified.
> > It should also be noted that I am not concerned by the reverse estimation (I(x, h_i)) as in this direction a model of identical capacity is retrained from scratch for all layers.

---

> > > ### Author Response · Authors · 2018-11-22
> > > **Response to training complexity and MI curves**
> > >
> > > Regarding the training complexity and the results you listed on PixelCNN++. This was for the CIFAR-10 dataset, but we're using the CINIC-10 dataset. The difference in size and challenges of this dataset means that those times are mostly irrelevant. We used CINIC-10 because we could split it three ways to avoid any data-contamination between training the models under scrutiny, training the decoder models, and evaluation of MI. Even after two weeks of training (which equates to 250 epochs of training on the 'validation' subset of CINIC-10) the PixelCNN++ models, convergence was still not reached and the measured losses were still decreasing. We chose to stop there even though the estimation could be better. That all said, given this dataset, 24 hours of training would equate to less than 20 epochs.
> > >
> > > More points on the curves would always be better, but the cost is simply too high for that. Our intention with this research was to apply MI tracking to this realistic scenario and in so doing add to the ongoing discussion on the information bottleneck.
> > >
> > > Regarding the question of figure 3a. Consider that retraining h_j (j > i) to estimate h_i, at a fixed point in the original training time (e.g., at some epoch) involves training those layers where j > i to convergence. h_j at that specific time will not necessarily be converged and when you estimate MI(y, h_j) it will have that layer as fixed/frozen. You are likely correct that h_j will throw the same information away when retraining as it did AT THE END of the original training run.
> > >
> > > The important nuance here is that when comparing MI(y, h_i) and MI(y, h_j) at some point before convergence (of the original ResNet classifier), the h_j is fixed to be at a sub-optimal point compared to where it eventually converges later. Therefore, when decoding to estimate MI(y, h_i), h_j is allowed to train and reaches convergence - a state that is different from when it itself is used to measure MI(y, h_j).

---

> > > > ### Comment · AnonReviewer2 · 2018-11-26
> > > > **Response to response to training complexity and MI curves**
> > > >
> > > >
> > > > Thank you for your answer.
> > > >
> > > >  > [We used CINIC-10 because we could split it three ways to avoid any data-contamination between training the models under scrutiny, training the decoder models, and evaluation of MI. Even after two weeks of training (which equates to 250 epochs of training on the 'validation' subset of CINIC-10) the PixelCNN++ models, convergence was still not reached and the measured losses were still decreasing. We chose to stop there even though the estimation could be better. That all said, given this dataset, 24 hours of training would equate to less than 20 epochs.]
> > > >
> > > >  Yes indeed, I wasn't thinking about the different dataset anymore, and I agree that this will incur longer training time. Given that the resolution is the same (32x32), and that the training split is of size 90k VS 50k for cifar10, it seems to be that the data regimes do not differ wildly. Seeing the images in the paper that introduces CINIC-10, it appears that images coming from Imagenet do have more variability, even once cropped, so this can also account for slower convergence. Just for the sake of certainty, I would like to know if the convolutional structure (number of layers, units per layer, downsampling...) has been kept the same as in pixelCNN++. I suppose that is what you mean by <<The hyper-parameters for the PixelCNN++ decoder models were set according to the original paper>>. If the architecture is changed, these details could be considered as an appendix in the paper. Since the convergence regime is key here, I think it would also help if you could provide training and test curves for the different pixCNN++ in the annex, with likelihood plotted against the number of epochs, to asses the quality of convergence.
> > > >
> > > >
> > > > > [Regarding the question of figure 3a. Consider that retraining h_j (j > i) to estimate h_i, at a fixed point in the original training time (e.g., at some epoch) involves training those layers where j > i to convergence. h_j at that specific time will not necessarily be converged and when you estimate MI(y, h_j) it will have that layer as fixed/frozen. You are likely correct that h_j will throw the same information away when retraining as it did AT THE END of the original training run.
> > > >
> > > > The important nuance here is that when comparing MI(y, h_i) and MI(y, h_j) at some point before convergence (of the original ResNet classifier), the h_j is fixed to be at a sub-optimal point compared to where it eventually converges later. Therefore, when decoding to estimate MI(y, h_i), h_j is allowed to train and reaches convergence - a state that is different from when it itself is used to measure MI(y, h_j).]
> > > >
> > > > Thank you for this clarification.

---

> > > > > ### Author Response · Authors · 2018-11-26
> > > > > **PixelCNN++ training curves and an unconditional PixelCNN++**
> > > > >
> > > > > Thank you very much for your response.
> > > > >
> > > > > We have updated the script to include training curves in the appendix, as you requested.
> > > > >
> > > > > We have also included training curves and samples for an unconditional PixelCNN++ trained on the encoder split of CINIC-10, in order to give better context to this work.
> > > > >
> > > > > You are correct in assuming that we have kept the hyper parameters, including architecture, etc. exactly the same as the original PixelCNN++ work.
> > > > >
> > > > > We hope that this clarifies things even more.

---

### Meta-Review · Area_Chair1 · 2018-12-13
**Neat approach, but more validation is needed**

**Confidence:** 4
**Recommendation:** Reject

**Metareview:**

This paper explores an approach to testing the information bottleneck hypothesis of deep learning, specifically the idea that layers in a deep model successively discard information about the input which is irrelevant to the task being performed by the model, in full-scale ResNet models that are too large to admit the more standard binning-based estimators used in other work. Instead, to lower-bound I(x;h), the authors propose using the log-likelihood of a generative model (PixelCNN++). They also attempt visualize what sort of information is lost and what is retained by examining PixelCNN++ reconstructions from the hidden representation at different positions in a ResNet trained to perform image classification on the CINIC-10 task. To lower-bound I(y;h), they perform classification. In the experiments, the evolution of the bounds on I(x;h) and I(y;h) are tracked as a function of training epoch, and visualizations (reconstructions of the input) are shown to support the argument that color-invariance and diversity of samples increases during the compression phase of training. These tests are done on models trained to perform either image classification or autoencoding. This paper enjoyed a good discussion between the reviewers and the authors. The reviewers liked the quantitative analysis of "usable information" using PixelCNN++, though R2 wanted additional experiments to better quantify the limitations of the PixelCNN++ model to provide the reader with a better understanding of plots in Fig. 3, as well as more points sampled during training. Both R2 and R3 had reservations about the qualitative analysis based on the visualizations, which constitute the bulk of the paper. Unfortunately, the PixelCNN++ training is computationally intensive enough that these requests could not be fulfilled during the ICLR discussion phase. While the AC recommends that this submission be rejected from ICLR, this is a promising line of research. The authors should address the constructive suggestions of R2 and R3 and submit this work elsewhere.